# Weak-shot Fine-grained Classification via Similarity Transfer

**Junjie Chen, Li Niu**,* **Liu Liu, Liqing Zhang**\*
MoE Key Lab of Artificial Intelligence,
Department of Computer Science and Engineering,
Shanghai Jiao Tong University
`{chen.bys, ustcnewly, shirlley}@sjtu.edu.cn, zhang-lq@cs.sjtu.edu.cn`

## Abstract

Recognizing fine-grained categories remains a challenging task, due to the subtle distinctions among different subordinate categories, which results in the need of abundant annotated samples. To alleviate the data-hungry problem, we consider the problem of learning novel categories from web data with the support of a clean set of base categories, which is referred to as weak-shot learning. In this setting, we propose a method called SimTrans to transfer pairwise semantic similarity from base categories to novel categories. Specifically, we firstly train a similarity net on clean data, and then leverage the transferred similarity to denoise web training data using two simple yet effective strategies. In addition, we apply adversarial loss on similarity net to enhance the transferability of similarity. Comprehensive experiments demonstrate the effectiveness of our weak-shot setting and our SimTrans method. Datasets and codes are available at https://github.com/bcmi/SimTrans-Weak-Shot-Classification.

## 1 Introduction

Deep learning methods have made a significant advance on extensive computer vision tasks. A large part of this advance has come from the available large-scale labeled datasets. For fine-grained classification, it is more necessary but more expensive to collect large-scale datasets. On the one hand, the subtle differences among fine-grained categories dramatically boost the demand for abundant samples. On the other hand, professional knowledge is usually required to annotate images for enormous subcategories belonging to one category. As a consequence, fine-grained classification is critically limited by the scarcity of well-labeled training images.

In practice, we often have a set of base categories with sufficient well-labeled data, and the problem is how to learn novel categories with less expense, in which base categories and novel categories have no overlap. Such problem motivates zero-shot learning [19], few-shot learning [6], as well as our setting. To bridge the gap between base (*resp.*, seen) categories and novel (*resp.*, unseen) categories, zero-shot learning requires category-level semantic representation (*e.g.*, word vector [27] or human annotated attributes [19]) for all categories, while few-shot learning requires a few clean examples (*e.g.*, 5, 10) for novel categories. Despite the great success of zero-shot learning and few-shot learning, they have the following drawbacks: 1) Annotating attributes or a few clean samples require expert knowledge, which is not always available; 2) Word vector is free, but much weaker than human annotated attributes [1]. Fortunately, large-scale images are freely available from public websites by using category names as queries, which is a promising data source to complement the learning of novel fine-grained categories without any manual annotation.

---

*Corresponding author

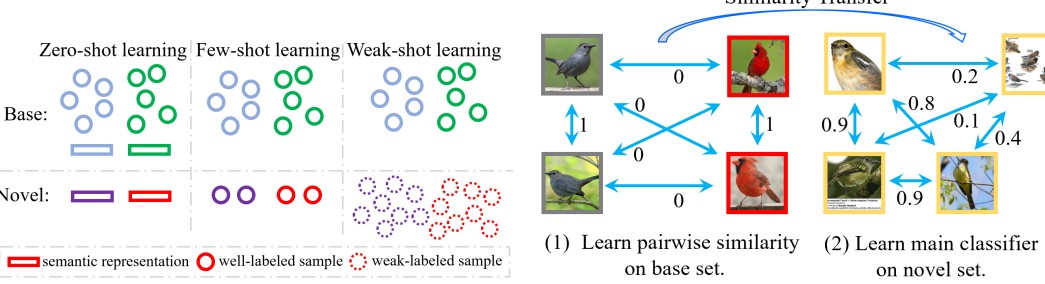

Figure 1: Comparison among zero-shot learning, few-shot learning, and weak-shot learning. Different colors indicate different categories.

Figure 2: We transfer the similarity learnt from base training set to enhance the main classifier learnt on novel training set. The boxes in different colors denote different categories. The numbers above the arrows indicate similarities.

Considering the drawbacks of zero/few-shot learning and the accessibility of free web data, we intend to learn novel categories by virtue of web data with the support of a clean set of base categories, which is referred to as weak-shot learning as illustrated in Figure 1. Formally, given a set of novel fine-grained categories which are not associated with any clean training images, we collect web images for novel categories as weak-labeled images and meanwhile leverage the clean images from base fine-grained categories. We refer to the clean (*resp.*, web) image set from base (*resp.*, novel) categories as base (*resp.*, novel) training set. Weak-shot learning is a useful and important setting. By taking car model classification as an example, we already have the dataset CompCars [52] with well-labeled images from base fine-grained categories, but we often need to recognize other fine-grained categories beyond this scope, because there are a huge number of car models and new car models are also continuously emerging. In this case, we could apply our setting to recognize novel fine-grained categories by collecting the web images for these categories. The closest related work to ours is [31], but they further assumed the reliability of word vectors [27] and the availability of unlabeled test images in the training stage.

In weak-shot setting, the key issue of novel training set is label noise, which will significantly degrade the performance of learnt classifier [31, 34]. We explore using base training set to denoise novel training set, although they have disjoint category sets. As illustrated in Figure 2, our proposed framework employs the pairwise semantic similarity to bridge the gap between base categories and novel categories. The pairwise similarity which denotes whether two images belong to the same category is category-agnostic, so it is highly transferable across category sets even if they are disjoint. Meanwhile, the pairwise similarity can be easily learnt from limited data, indicating that a small set of already annotated images could help learn extensive novel categories (see Section 5.3). Analogously, some methods [3] for few-shot learning transferred similarity from base categories to novel categories, which is directly used for classification. In contrast, we transfer the pairwise similarity to alleviate the label noise issue of web data. For learning from web data, some works [38, 11] also attempted to denoise by similarity. Nevertheless, their similarities are derived from noisy samples, and likely to be corrupted due to noise overfitting [17].

Specifically, our framework consists of two training phases. Firstly, we train a similarity net (SimNet) [54] on base training set, which takes in two images and outputs the semantic similarity. Secondly, we apply the trained SimNet to obtain the semantic similarities among web images. In this way, the similarity is transferred from base categories to novel categories. Based on the transferred similarities, we design two simple yet effective methods to assist in learning the main classifier on novel training set. 1) Sample weighting (*i.e.*, assign small weights to the images dissimilar to others) reduces the impact of outliers (web images with incorrect labels) and thus alleviates the problem of noise overfitting. 2) Graph regularization (*i.e.*, pull close the features of semantically similar samples [60]) prevents the feature space from being disturbed by noisy labels. In addition, we propose to apply adversarial loss [9] on SimNet to make it indistinguishable for base categories and novel categories, so that the transferability of similarity is enhanced. Since the key of our method is similarity transfer, we name our method *SimTrans*. We conduct extensive experiments on three fine-grained datasets to demonstrate that the pairwise similarity is highly transferable and dramatically

benefits learning from web data, even when the category sets are disjoint. Although the focus of this paper is fine-grained classification, we also explore the effectiveness of our setting and method on a coarse-grained dataset ImageNet [5]. We summarize our contributions as

- We propose to use transferred similarity to denoise web training data in weak-shot classification task, which has never been explored before.

- We propose two simple yet effective methods using transferred similarity to tackle label noise: sample weighting and graph regularization.

- One minor contribution is applying adversarial loss to similarity net to enhance the transferability of similarity.

- Extensive experiments on four datasets demonstrate the practicality of our weak-shot setting and the effectiveness of our SimTrans method.

## 2 Related Work

### 2.1 Zero-shot and Few-shot Learning

Zero-shot learning employs category-level semantic representation (*e.g.*, word vector or annotated attributes) to bridge the gap between seen (*resp.*, base) categories and unseen (*resp.*, novel) categories. A large part of works [2, 8, 32, 50, 10] learn a mapping between visual features and category-level semantic representations. Our learning scenario is closer to few-shot learning.

Few-shot learning depends on a few clean images (*e.g.*, 5-shot or 10-shot) to learn each novel category, which could be roughly categorized as the following three types. Optimization-based methods [7, 36] optimize the classifier on a variety of learning tasks (*e.g.*, learn each category by a few images), such that it can solve new learning tasks using only a small number of images (*e.g.*, learn novel categories with a few images). Memory-based methods [37, 35, 28] employ memory architectures to store key training images or directly encode fast adaptation algorithms. Metric-based methods [39, 42, 33, 3] learn a deep representation with a similarity metric in feature space and classify test images in a nearest neighbors manner.

Concerning the problem setting, both zero-shot learning and few-shot learning ignore the freely available web images, whereas we learn novel categories by collecting web images using category names as queries. Concerning the technical solution, metric-based few-shot learning methods mainly learn image-category similarities and directly recognize test images according to the similarity. In contrast, we transfer image-image similarities to reveal the semantic relationships among web training images, which are used to denoise the web data for a better classifier.

### 2.2 Webly Supervised Learning

Due to the data-hungry property of deep learning, learning from web data has attracted increasing attention. Many methods have been proposed to deal with noisy images by outlier removal [24, 49], robust loss function [29, 45, 57], label correction [34, 43, 46], multiple instance learning [61, 56], and so on [55, 41, 53, 16, 30]. A prevalent research direction closely related to ours is dealing with label noise using similarities. To name a few, CurriculumNet [38] computed Euclidean distances between image features, and then designed curriculums according to the distances. SelfLearning [11] employed the cosine similarity between image features to select category prototypes and correct labels. SOMNet [44] leveraged self-organizing memory module to construct similarity between image and category, which could simultaneously tackle label noise and background noise of web images. However, the similarities used in above methods are derived from the noisy training set, which are likely to be corrupted by noise and lead to sub-optimal results [17, 51]. To alleviate this problem, recent works [17, 51] introduced additional annotations to correct label noise. For example, CleanNet [17] directly learned the image-category similarity based on verification labels, which involves human verification of noisy images. Distinctive from the above methods, we do not require any manual annotation on crawled web images.

### 2.3 Weak-shot Learning

In broad sense, weak-shot learning means learning novel categories from cheap weak labels with the support of a set of base categories already having strong labels. Similar weak-shot setting has been explored in other computer vision tasks, including object detection [13, 25, 58, 4, 22], semantic segmentation [59], and instance segmentation [14, 18]. For weak-shot object detection (also named mixed-supervised or cross-supervised), base categories have bounding box annotations while novel categories only have image class labels. For weak-shot semantic segmentation, base categories have semantic mask annotations while novel categories only have image class labels. For weak-shot instance segmentation (also named partially-supervised), base categories have instance mask annotations while novel categories only have bounding box annotations. To the best of our knowledge, we are the first work on weak-shot classification, which employs transferred similarity to denoise web training data.

## 3 Problem Definition

In the training stage, we have $C_b$ base categories with $N_c^b$ clean images for the $c$-th base category. Our main goal is learning to classify $C_n$ novel categories given $N_c^n$ web training images for the $c$-th novel category. Base categories and novel categories have no overlap. No extra information (*e.g.*, word vector) is required in the training stage. In the testing stage, test images come from novel categories. We refer to the above setting as weak-shot learning.

For some real-world applications, test images may come from both base categories and novel categories, which is referred to as generalized weak-shot learning. In this paper, we focus on weak-shot learning, while leaving generalized weak-shot learning to Appendix.

## 4 Approach

Our training stage consists of two training phases: learning similarity net (SimNet) on base training set and learning the main classifier on novel training set.

### 4.1 Learning SimNet on Base Training Set

Although there are many forms of similarity, we choose an end-to-end deep network SimNet to model the similarity function as in [54]. The architecture of SimNet is shown in the top pipeline of Figure 3, which takes in a mini-batch of images and outputs the pairwise similarity/dissimilarity score for each pair of input images (*e.g.*, $M^2$ pairs for mini-batch size $M$). The enumeration (Enum) layer simply concatenates each pair of image features which are extracted by the backbone. For instance, suppose the image features have size $M \times D$

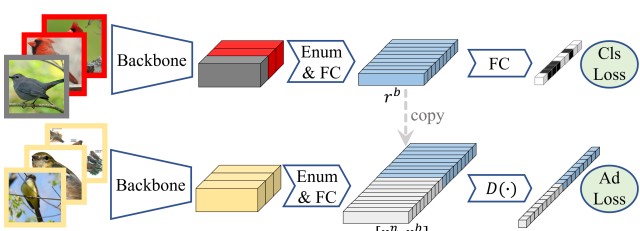

Figure 3: Illustration of similarity net with adversarial loss. The top (*resp.*, bottom) pipeline processes base (*resp.*, novel) training set. $\mathbf{r}^b$ (*resp.*, $\mathbf{r}^n$) represents base (*resp.*, novel) relation features.

with the feature dimension $D$, then the output of the enumeration layer will have size $M^2 \times 2D$. After that, one fully-connected (FC) layer extracts feature for each concatenated feature pair, which is dubbed as relation feature $\mathbf{r}$ of each pair. Finally, the relation features are supervised by classification loss with binary labels being "similar" (a pair of images from the same category) and "dissimilar" (a pair of images from different categories).

Considering the training and testing for SimNet, if we construct the mini-batch randomly, similar and dissimilar pairs will be dramatically imbalanced. Specifically, if there are $C$ categories in a mini-batch, the probability for two images to be from the same category is $\frac{1}{C}$. To reduce the imbalance between similar pairs and dissimilar pairs, when constructing a mini-batch, we first randomly select

$C_m$ ($\ll C$) categories and then randomly select $\frac{M}{C_m}$ images from each selected category, as in [54]. We use $C_m = 10$ and $M = 100$ for both training and testing of SimNet.

#### 4.1.1 Adversarial Loss

Ideally, the learnt similarity is category-agnostic, but there may exist domain gap between the relation features of different category sets. Therefore, we use novel training set in the training stage to further reduce the domain gap between base categories and novel categories, as shown in the bottom pipeline of Figure 3.

Specifically, a discriminator $D(\cdot)$ takes in relation features $\mathbf{r}$ in SimNet and recognizes whether they come from base categories ($\mathbf{r}^b$) or novel categories ($\mathbf{r}^n$). The SimNet acts as generator, aiming to produce relation features which could not only confuse the discriminator but also benefit the relation classification. Note that the labels of novel training images are noisy, so we exclude the image pairs from novel categories from the relation classification loss. Analogous to [9], we optimize SimNet and the discriminator alternatingly in an adversarial manner. Firstly, we freeze the generator and minimize the adversarial loss of discriminator. Secondly, with frozen discriminator, we minimize the relation classification loss of SimNet and maximize the adversarial loss of discriminator. The classification loss and the adversarial loss are balanced with a hyper-parameter $\beta$, set as $0.1$ via cross-validation. The optimizing is trivial and we leave the details to Appendix.

### 4.2 Learning Classifier on Novel Training Set

Because of the label noise of web images, the performance will be significantly degraded when directly training the classifier using web data. To address the label noise issue, we employ two simple yet effective methods based on transferred similarities as illustrated in Figure 4. Transferred similarities mean the similarities among novel training samples, which are calculated by SimNet trained on base training set. Next, we will introduce these two methods, *i.e.*, sample weighting and graph regularization, separately.

#### 4.2.1 Sample Weighting

For the web images within a novel category, we observe that non-outliers (images with the correct label) are usually dominant, while outliers (images with incorrect labels) are from non-dominant inaccurate categories. When calculating the semantic similarities among web images within a novel category, outliers are more likely to be dissimilar to most other images. Therefore, we could determine whether an image is an outlier according to its similarities to other images.

Formally, for the $c$-th novel category with $N_c^n$ web images, we first compute the similarity matrix $\mathbf{S}_c \in \mathbb{R}^{N_c^n \times N_c^n}$, with each entry $s_{c,i,j}$ being pairwise similarity calculated by SimNet pre-trained on base training set. Although the size of $\mathbf{S}_c$ may be large, there are only $N_c^n$ times of backbone inference and $N_c^n \times N_c^n$ times of two FCs inference, which are computationally efficient. Then, we employ the average of similarities be-

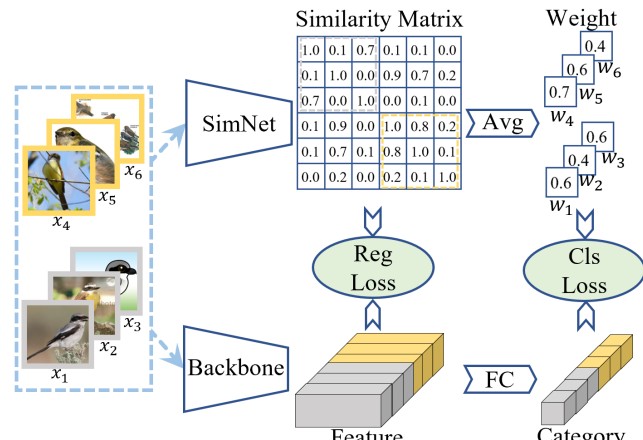

Figure 4: The overview of learning novel categories from web data. The similarity net (SimNet) outputs similarity matrix with pairwise similarities and generates sample weights. The sample weights are employed to weigh the main classification loss (Cls Loss) of each image, while the similarity matrix is used in graph regularization loss (Reg Loss) based on image features.

tween image $i$ and all the other images as its weight:

$$w_{c,i} = \frac{1}{N_c^n} \sum_{j=1}^{N_c^n} \frac{s_{c,i,j} + s_{c,j,i}}{2}. \tag{1}$$

Then, the sample weights are normalized to have unit mean, *i.e.*, $\bar{w}_{c,i} = \frac{w_{c,i}}{\sum_{j=1}^{N_c^n} w_{c,j}/N_c^n}$. As analysed before, samples with lower weights $\bar{w}_{c,i}$ are more likely to be outliers. Finally, we employ weighted classification loss based on sample weights $\bar{w}_{c,i}$. In this way, we assign lower weights to the training losses of outliers, which enables non-outliers to contribute more to learning a robust classifier:

$$L_{cls\_w} = \frac{1}{M} \sum_{m=1}^{M} -\bar{w}_m \log f(\mathbf{x}_m)_{y_m}, \tag{2}$$

where $M$ is the mini-batch size, $\mathbf{x}_m$ is the $m$-th image in the mini-batch, $\bar{w}_m$ is its assigned weight, and $f(\mathbf{x}_m)_{y_m}$ is its classification score corresponding to its category $y_m$.

### 4.2.2 Graph Regularization

When directly learning on novel training set, the feature graph structure, that is, the similarities among image features, are determined by noisy labels. In particular, the classification loss implicitly pulls features close for images with the same labels. However, the feature graph structure may be misled by noisy labels, so we attempt to use transferred similarities to rectify the misled feature graph structure. Specifically, we employ typical graph regularization [60] based on transferred similarities to regulate features, which enforces the features of semantically similar images to be close.

Formally, for each mini-batch of $M$ images, we first compute the similarity matrix $\tilde{\mathbf{S}} \in \mathbb{R}^{M \times M}$ using the SimNet pre-trained on base training set. Regarding the similarity matrix as adjacency matrix, the graph regularization loss is formulated as:

$$L_{reg} = \sum_{i,j} \tilde{s}_{i,j} \|h(\mathbf{x}_i) - h(\mathbf{x}_j)\|_2^2, \tag{3}$$

where $\tilde{s}_{i,j}$ is each entry in $\tilde{\mathbf{S}}$, and $h(\mathbf{x}_i)$ is the image feature of $\mathbf{x}_i$ extracted by the backbone.

According to [40], web images mainly have two types of noise: outlier noise and label-flip noise. Outlier noise means that an image does not belong to any category within the given category set, and label-flip noise means that an image is erroneously labeled as another category within the given category set. For the samples with label-flip noise, sample weighting directly discards these samples by assigning lower weights. However, graph regularization can utilize them to maintain reasonable feature graph structure and facilitate feature learning, which could complement sample weighting.

### 4.3 The Full Objective

On the whole, we train the classifier on novel training set by minimizing the weighted classification loss and graph regularization loss:

$$L_{full} = L_{cls\_w} + \alpha L_{reg}, \tag{4}$$

where $\alpha$ is a hyper-parameter set as $0.1$ by cross-validation.

## 5 Experiments

### 5.1 Datasets and Implementation

We conduct experiments based on three fine-grained datasets: CompCars [52] (Car for short), CUB [48], and FGVC [26]. We split all categories into base categories and novel categories. The base training/test set and the novel test set are from the original dataset while the novel training set is constructed using web images. For instance, the novel training set of Car dataset is constructed using the released web images in WebCars [61], while the novel training sets of CUB and FGVC are constructed by collecting the web images by ourselves. The statistics of three datasets are summarized

Table 1: The statistics of splits on three datasets. **Category** shows the number of split categories. **Train** (*resp.*, **Test**) indicates the average number of training (*resp.*, test) images for each category.

| Dataset | Split | Category | Train | Test |
|---------|-------|----------|-------|------|
| Car | Base | 323 | 37 | 34 |
| | Novel | 108 | 510 | 36 |
| CUB | Base | 150 | 30 | 30 |
| | Novel | 50 | 1000 | 30 |
| FGVC | Base | 75 | 67 | 33 |
| | Novel | 25 | 1000 | 33 |

Table 2: Module contributions on CUB dataset. **Cls** indicates the cross-entropy classification loss of main classifier. **Ad** means training Sim-Net with adversarial loss. **Weight** means sample weighting and **Reg** means graph regularization.

| Cls | Ad | Weight | Reg | Acc (%) |
|-----|-----|--------|-----|---------|
| √ | | | | 85.4 |
| √ | | √ | | 90.3 |
| √ | | | √ | 89.5 |
| √ | | √ | √ | 91.2 |
| √ | √ | √ | | 90.7 |
| √ | √ | | √ | 90.1 |
| √ | √ | √ | √ | 91.7 |

Table 3: The first three rows show the performances (%) of SimNet evaluated by 150 base categories, 50 base categories, and 50 novel categories. The last row shows the performance (%) of uniform random guess on 50 novel categories. **PR**, **RR**, and **F1** represent precision rate, recall rate, and F1-score, respectively.

| | *Similar* class | | | *Dissimilar* class | | |
|-------|------|------|------|------|------|------|
| | PR | RR | F1 | PR | RR | F1 |
| B-150 | 84.4 | 87.7 | 86.0 | 98.6 | 98.2 | 98.4 |
| B-50 | 89.0 | 88.2 | 88.6 | 98.7 | 99.0 | 98.8 |
| N-50 | 88.4 | 87.6 | 88.0 | 98.6 | 98.7 | 98.7 |
| Rand* | 10.0 | 50.0 | 16.7 | 90.0 | 50.0 | 64.3 |

Table 4: The performances (%) of main classifier supported by various similarities learnt from various training sets.

| Source | Type | Classifier |
|--------|------|-----------|
| Novel | Euclidean | 86.2 |
| | Cosine | 86.4 |
| | SimNet | 87.6 |
| Novel+ Base | Euclidean | 87.5 |
| | Cosine | 87.6 |
| | SimNet | 89.5 |
| Base | Euclidean | 88.8 |
| | Cosine | 89.1 |
| | SimNet | 91.7 |

in Table 1. We employ ResNet50 [12] pretrained on ImageNet [5] as our backbone. There is no overlap between novel categories and ImageNet $1k$ categories. Besides, all baselines use the same backbone for a fair comparison. More details of datasets and implementation are left to Appendix.

Note that the focus of this work is fine-grained classification, because 1) it especially requires expert knowledge to annotate labels for fine-grained data (a drawback of few-shot learning); 2) we conjecture that semantic similarity should be more transferable across fine-grained categories. However, we also explore the effectiveness of our setting and method on a coarse-grained dataset ImageNet [5]. We leave the experimental results on ImageNet to Appendix.

## 5.2 Ablation Study

We conduct our ablation study on CUB dataset, considering its prevalence in extensive vision tasks. We evaluate the performances of different combinations of our modules, and summarize the results in Table 2. Solely using sample weighting leads to a dramatic improvement (90.3% *v.s.* 85.4%). Solely enabling the graph regularization also results in a considerable advance (89.5% *v.s.* 85.4%). Jointly applying sample weights and graph regularization further improves the performance (91.2% *v.s.* 85.4%). The adversarial loss on SimNet boosts the performance of classifier for both sample weighting (90.7% *v.s.* 90.3%) and graph regularization (90.1% *v.s.* 89.5%). Finally, our full method outperforms the baseline by a large margin (91.7% *v.s.* 85.4%).

## 5.3 Investigating Similarity Transfer

We evaluate SimNet in various situations to explore the effectiveness of similarity transfer. Although SimNet is employed to denoise web images, we cannot evaluate its performance on noisy images without ground-truth labels, so we perform evaluation on clean test images. The performance measurements (*e.g.*, precision rate (PR), recall rate (RR), and F1-score (F1)) are computed at the pair level, instead of image level. The experiments in this subsection are conducted on CUB dataset.

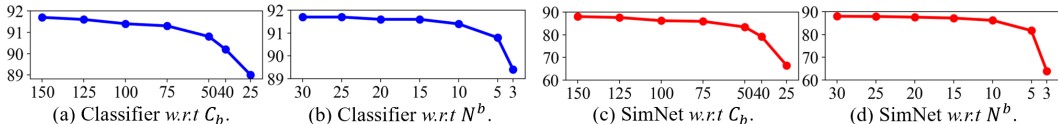

Figure 5: Performance variations of main classifier (accuracy %) or similarity net (F1-score %) *w.r.t* different numbers of used base categories ($C_b$) or used clean images per base category ($N^b$).

### 5.3.1 The Transferability of Similarity

The performance gap of SimNet between base categories and novel categories indicates the transferability of learnt similarity. We first train SimNet on base training set with 150 categories, and evaluate its performances on base test set with 150 categories and novel test set with 50 categories. We also evaluate on the base test set formed by 50 random base categories to exclude the influence of different category numbers. The results are summarized in Table 3. Surprisingly, SimNet trained on base categories achieves a comparable performance for base categories and novel categories (Base-50 *v.s.* Novel-50). On the whole, the above results suggest that the pairwise similarities are highly transferable across fine-grained categories.

### 5.3.2 The Comparison of Similarity Sources and Types

To demonstrate the superiority of the our transferred similarity, we compare different types of similarities learnt from various training sets (sources) in Table 4. For *Euclidean* and *Cosine*, pairwise similarity is computed based on the features extracted by a network pretrained on a specific training set (source). *Cosine* similarity is well-known and we omit the details here. For *Euclidean*, we adopt the reciprocal of Euclidean distance. For SimNet, we can train it using different training sets (sources). By comparing different similarity sources, we observe that all types of similarities learned from base training set perform better, while the performances decrease when novel training set (containing noisy images) is involved. This verifies that the similarity could be severely corrupted by noisy images, indicating the necessity of learning similarity using clean images even if the category sets are disjoint. By comparing different similarity types, SimNet performs optimally, showing the superiority of learning similarity in an end-to-end fashion.

### 5.3.3 The Impact of the Scale of Base Training Set

We naturally wonder what is the minimum requirement of the scale of base training set for our similarity net and main classifier to perform well. We explore the scale of base training set from two aspects: category number and image number in each category. We mainly report the F1-score of *similar* category evaluated on novel test set as the performance of SimNet, and the results are illustrated in Figure 5. The subfigure (a) and (c) shows the robustness when using more than 50 categories, while the subfigure (b) and (d) shows robustness when using more than 5 images per category. Furthermore, we explore the joint impact of two aspects in Table 5. Surprisingly, supported by only 50 base categories with 5 images in each category, the main classifier could achieve a satisfactory performance, *i.e.*, 89.3% against 85.4%, where 85.4% is the result without the support of base training set. This study indicates the potential of our setting and method, that is, a small-scale off-the-shelf base training set could facilitate learning extensive novel fine-grained categories.

## 5.4 Comparison with Prior Works

We compare with two types of methods: webly supervised learning and transfer learning.

For webly supervised learning, we compare with recent representative methods: SelfLearning [11], MetaCleaner [56], CurriculumNet [38], SOMNet [44], DivideMix [21], LearnToLearn [20], NLNL [15], and JoCoR [47]. Webly supervised learning methods only use novel training set.

For transfer learning across categories, there are mainly two groups of methods: few-shot learning and knowledge transfer. For few-shot learning, we compare with MetaBaseline [3], MAML [7], and SNAIL [28]. MetaBaseline is a metric-based method, and we use the averaged feature of each novel category as class centroid following [3]. SNAIL is a memory-based method, and we use 16 prototypes of each novel category as 16 shots due to hardware limitation. For knowledge transfer,

Table 5: The performances (%) of SimNet and main classifier supported by various scales $C \times N$ of base training set, in which $C$ denotes the used category number and $N$ denotes the used image number in each category.

| $C \times N$ | $50 \times 5$ | $75 \times 5$ | $50 \times 10$ | $150 \times 30$ |
|---|---|---|---|---|
| SimNet | 73.2 | 79.1 | 80.6 | 88.0 |
| Classifier | 89.3 | 89.9 | 90.4 | 91.7 |

Table 6: Accuracies (%) of representative methods using various numbers of web images on CUB dataset.

| Number | 200 | 400 | 600 | 800 | 1000 |
|---|---|---|---|---|---|
| Cls Loss | 78.5 | 82.7 | 84.3 | 85.2 | 85.4 |
| Fusion1 | 80.7 | 84.9 | 87.1 | 88.2 | 88.3 |
| Fusion2 | 81.0 | 85.1 | 87.2 | 88.3 | 88.5 |
| **Ours** | **84.3** | **89.5** | **91.0** | **91.6** | **91.7** |

Table 7: Accuracies (%) of various methods on three datasets in the weak-shot learning. The best results are highlighted in boldface.

| Method | Car | CUB | FGVC |
|---|---|---|---|
| Cls Loss | 83.1 | 85.4 | 86.6 |
| SelfLearning | 85.1 | 87.3 | 88.1 |
| MetaCleaner | 84.9 | 87.1 | 88.3 |
| CurriculumNet | 85.2 | 86.8 | 87.9 |
| SOMNet | 86.0 | 87.9 | 88.6 |
| DivideMix | 86.2 | 88.0 | 89.1 |
| LearnToLearn | 85.3 | 87.6 | 88.4 |
| NLNL | 85.4 | 87.9 | 88.2 |
| JoCoR | 85.9 | 87.9 | 88.3 |
| MetaBaseline | 85.8 | 87.1 | 88.7 |
| MAML | 84.6 | 86.8 | 87.9 |
| SNAIL | 84.1 | 86.3 | 87.2 |
| Distillation | 83.7 | 85.9 | 87.3 |
| Finetune | 84.2 | 86.5 | 87.6 |
| Fusion1 | 86.8 | 88.3 | 89.7 |
| Fusion2 | 86.9 | 88.5 | 90.0 |
| SimTrans | **89.8** | **91.7** | **92.8** |

the *Distillation* baseline [23] enhances the classifier for novel categories by additionally predicting category distribution over base categories, which is supervised by a classifier pre-trained on base training set. Another natural baseline, named as *Finetune*, trains the model on the mixture of base training set and novel training set, and then fine-tunes on novel training set.

In addition, as far as we are concerned, there is no method which could jointly denoise web training data and transfer across categories without additional information (*e.g.*, word vector). So we combine competitive methods in their own tracks using late fusion (*i.e.*, average prediction scores). The fusion "SOMNet+MetaBaseline" is referred to as *Fusion1*, while the fusion "DivideMix+MetaBaseline" is referred to as *Fusion2*.

All the results are summarized in Table 7. We also include the basic baseline *Cls Loss*, which is trained on novel training set with standard classification loss (row 1 in Table 2). Based on Table 7, we have the following observations:

1) Webly supervised learning methods outperform the basic baseline *Cls Loss*, showing the general improvement brought by denoising web data without any support.

2) Few-shot learning methods and knowledge transfer methods outperform the basic baseline *Cls Loss*, indicating the advantage of transfer learning across categories. Note that *MetaBaseline* achieves a commendable performance, probably because averaging features could denoise web images to some extent. Nevertheless, the overall performances of transfer learning methods are limited by severe label noise.

3) The simple combination of webly supervised learning and transfer learning, *i.e.*, *Fusion1* or *Fusion2*, outperforms the above baselines, which directly demonstrates the effectiveness of our weak-shot setting. Furthermore, our method achieves the optimal performance against all baselines, which indicates the superiority of sample weighting and graph regularization based on similarity transfer.

## 5.5  Qualitative Analysis for Weight and Graph

We visualize to verify whether the higher weights are assigned to clean images and the transferred similarities could reveal semantic relations. The visualizations and analyses are left to Appendix.

## 5.6  Robustness Analysis of The Method

We analyse the robustness of our method to several factors, including the number of web images, the selection of hyper-parameters, the noise ratio in web images, and the impact of different backbones.

Table 8: Accuracies (%) of different methods on three datasets in the generalized weak-shot setting. The best results are highlighted in boldface.

| Method | Car | CUB | FGVC |
|---|---|---|---|
| Cls Loss | 76.3 | 73.3 | 76.9 |
| Fusion1 | 77.5 | 74.8 | 78.8 |
| Fusion2 | 78.1 | 75.0 | 79.3 |
| SimTrans | **80.4** | **76.9** | **81.7** |

For the impact of the number of web images, in our setting, the training data consists of clean images for base categories and web images for novel categories. The impact of the former is investigated in Section 5.3.3, and here we conduct experiments using different numbers of web images for each novel category on CUB dataset. As shown in Table 6, the general performances are gradually saturated given more training images (*e.g.*, more than 800 web images), and the observations of performance promotions against baselines are consistent across different numbers of web images.

For other factors (*i.e.*, hyper-parameter, noise ratio, and backbones), we left the analysis to Appendix considering the space limitation.

### 5.7 Extension to Generalized Setting

In the generalized weak-shot learning, we additionally include base test set from base categories in the test set, so that test images may come from a mixture of base categories and novel categories, which is more practical in real-world applications. We extend our method and baselines to generalized weak-shot learning scenario for comparison.

For simplicity, we directly conduct the experiments by treating the base training set as web images with low noise rate. For our method, the first training stage of training SimNet remains the same, while the second training stage additionally includes base training set when training the main classifier. Thus, the sample weights assigned to base training images would be near one, and the graph regularization involving base training images will also do no harm to feature learning at least.

We compare with the basic baseline *Cls Loss* as well as two combinations of webly supervised learning method and transfer learning method across categories. SOMNet and DivideMix are also conducted by treating base training images as web images (the clean probability thresholds are set as 0 for base training images in DivideMix). MetaBaseline could be directly applied in the generalized setting. One practical problem is that the image number of base categories and the image number of novel categories are highly imbalanced (*e.g.*, 30 and 1000 for CUB), so we weigh the classification loss of each category using their image numbers (higher weight for the category with fewer images) for all methods in this setting. The results are reported in Table 8, from which we can find that the fusion of transfer learning and webly supervised learning outperforms the basic baseline, while our method further improves the results by a large margin.

## 6 Conclusion

In this paper, we consider the problem of learning novel categories from easily available web images with the support of a set of base categories with off-the-shelf well-labeled images. Specifically, we have proposed to employ the pairwise semantic similarity to bridge the gap between base categories and novel categories. Based on the transferred similarities, two simple yet effective methods have been proposed to deal with label noise when learning the classifier on web data. Adversarial loss is also applied to enhance the transferability of similarity. Extensive experiments on four datasets have demonstrated the potential of our learning paradigm and the effectiveness of our method.

## Acknowledgements

The work was supported by the National Key R&D Program of China (2018AAA0100704), National Natural Science Foundation of China (Grant No. 61902247), the Shanghai Municipal Science and Technology Major Project (Grant No. 2021SHZDZX0102) and the Shanghai Municipal Science and Technology Key Project (Grant No. 20511100300).

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
