# OpenReview forum: "Weak-shot Fine-grained Classification via Similarity Transfer"
_NeurIPS.cc/2021/Conference — NeurIPS 2021 Poster_

### Official Review · Reviewer_wyya · 2021-07-15

**Rating:** 4
**Confidence:** 4

**Summary:**

This paper proposes to use transferred similarity to denoise web training data in weak-shot fine-grained classification task, with two simple yet effective strategies methods, sample weighting and graph regularization.


**Ethical Concerns:**

I do not think there are ethical issues with this paper.



**Limitations And Societal Impact:**

Please refer to the section "Main Review".



**Main Review:**

1. The novelty and contribution of this work are marginal. Although the employment of transferred similarity to denoise web training data in weak-shot fine-grained classification task is new, as claimed in the paper, the combination of SimNet and DANN (Domain Adversarial Neural Networks) fails to surprise me. It seems that the proposed sample weighting and graph regularization share similar idea and intuition with MADA (Multi-adversarial Domain Adaptation), making the technical novelty not quite significant.
2. The organization of the manuscript could be improved. I think it is not a reader-friendly choice to leave too many detailed results and analyses to Appendix (Section 5.8, Section 5.9, and Section 5.10), which may make the flow of the paper hard to follow.



**Time Spent Reviewing:**

5

---

> ### Author Response · Authors · 2021-08-10
> **Answer to Reviewer #4.**
>
> **Q1**. The novelty and contribution of this work are marginal. It seems that the proposed sample weighting and graph regularization share similar idea and intuition with MADA (Multi-adversarial Domain Adaptation), making the technical novelty not quite significant.
>
> **A1**. As summarized in Line 72-77, our major novelty is using transferred similarity to denoise web training data in the weak-shot setting, which has never been explored before. As for the technical solution, the graph regularization is not applied in MADA. Generally speaking, although the sample weighting and graph regularization have been widely used in previous literature, the key problem is how to use them in different tasks. We design our sample weighting and graph regularization based on the proposed cross-category similarity transfer, which is distinctive from previous works and well-tailored for the new weak-shot setting. Furthermore, since this is the first method in this new setting, we believe that it is not a drawback to make the proposed method simple and effective.

---

> > ### Comment · Reviewer_wyya · 2021-09-02
> > **Reply to authors**
> >
> > After reading the authors' rebuttal, my key concern of novelty has not been addressed. And I tend to remain my score of reject unchanged.

---

### Official Review · Reviewer_K9m3 · 2021-07-16

**Rating:** 5
**Confidence:** 5

**Summary:**

This paper proposes to transfer pairwise semantic similarity from clean image data to web image data for classifying novel fine-grained categories. The proposed method provides a way to alleviate the data-hungry problem in the fine-grained classification task.

**Limitations And Societal Impact:**

Yes.

**Main Review:**

This paper proposes to utilize the easily available web data to learn novel fine-grained categories by transferring the pairwise semantic similarity from clean data to web data. Experiments are conducted for effectiveness validation on three public datasets. My concerns are as follows:

1.	This paper holds the premise that most web images are with correct labels and utilize the proposed SimNet trained on clean data to filter out the noisy web data. Dose the proposed method still work when the crawled web data are very dirty? Corresponding experiments or theoretical analyses should be added.
2.	When the training set is composed of the base training set and the novel training set, the classification accuracy drops a lot in section 5.10. Respective classification results in the base test set and the novel test set should be given. Corresponding experiments and analyses should be added, which is essential for verifying the effectiveness of the proposed method in the real-world applications.
3.	The authors should describe the difference between weak-shot learning and zero/few-shot learning more clearly. Respective training and test stages should be presented for better understanding of readers.
4.	More qualitative results about finding the outliers among the web data should be added to prove the effectiveness of the proposed method visually.
5.	The layout of this paper should be carefully designed. For example, Figure 3, Figure 4 and their surrounding text should be reset for better readability.


**Time Spent Reviewing:**

8

---

> ### Author Response · Authors · 2021-08-10
> **Answer to Reviewer #3.**
>
> **Q1**. Does the proposed method still work when the crawled web data are very dirty?
>
> **A1**. Actually, we have no image labels for novel classes from web search and thus we cannot examine or control the noisy level for each class in the real-world applications. Nevertheless, in Section "5.8 The Impact of the Noise Ratio in Web Images", we have explored the impact of the noise ratio using synthetic noisy data to show the robustness of our setting and method.
>
> ---
>
> **Q2**. When the training set is composed of the base training set and the novel training set, the classification accuracy drops a lot in section 5.10 (the generalized setting). Respective classification results in the base test set and the novel test set should be given.
>
> **A2**. We show the detailed results in Tab.1 below, which are overall consistent with those in section 5.10, that is, the fusion of transfer learning and webly supervised learning outperforms the basic baseline for both test sets, while our method further improves the results dramatically for both test sets. Specifically, novel categories are improved significantly and base categories are promoted moderately. Although the base categories are associated with clean training data, their performance could also be suppressed by the noisy training data of novel categories (e.g., label flip noise). Thanks to the sample weighting and graph regularization, our proposed method show superior performances on both base test set and novel test set.
>
> As for the performance drops of generalized setting against the typical setting, the overall task becomes much more difficult, because of a larger number of fine-grained categories (e.g., 50 v.s. 200 for CUB dataset) and data imbalance issue (e.g., 30 images per base category v.s. 1000 images per novel category for CUB dataset as described in Line 121-123 of Appendix). Nevertheless, we explore using an intuitive adaption for all methods in section 5.10 (with the class weighting strategy to the data imbalance issue) and have demonstrated the effectiveness of the proposed method against baselines. As claimed in Line 122-123, we focus on the typical setting in this paper, and we would like to strengthen the solution (e.g., to better solve the imbalance issue) for the generalized setting in the future work.
>
> **Tab 1**. Accuracies (%) of different methods on base and novel test set in the generalized setting. "Fusion1" and "Fusion2" are the short names for "SOMNet+MetaBaseline" and "DivideMix+MetaBaseline", which are the two most competitive baselines as descriped in Line 312-314.
>
> | Method   | Car-Base | Car-Novel | CUB-Base | CUB-Novel | FGVC-Base | FGVC-Novel |
>  :-: | :-: | :-:| :-:| :-: | :-:| :-:
> | Cls Loss | 80.9     | 63.2      | 76.3     | 64.4      | 77.7      | 74.3       |
> | Fusion1  | 81.2     | 67.1      | 76.9     | 68.7      | 78.9      | 78.4       |
> | Fusion2  | 81.9     | 67.3      | 77.0     | 69.2      | 79.3      | 79.2       |
> | Ours     | 84.1     | 69.8      | 78.4     | 72.1      | 81.1      | 83.5       |
> ---
>
> **Q3**. Describe the difference between weak-shot learning and zero/few-shot learning more clearly. Respective training and test stages should be presented for better understanding of readers.
>
> **A3**. The difference in data setting is shown in Fig. 1 and described in Line 23-31, zero-shot learning setting usually learns from clean base images and semantic base/novel word vectors; few-shot learning setting usually learns from clean base images and a few clean novel images; and weak-shot learning learns from clean base images and noisy novel images. The difference in data setting inherently determines that they have different training and test stages. Moreover, different groups of methods in one field (e.g., few-shot learning setting) can also have different training and test stages. Since zero/few-shot learning is not the focus of this paper, we omit the details here.

---

### Official Review · Reviewer_hoaz · 2021-07-16

**Rating:** 5
**Confidence:** 4

**Summary:**

This paper addresses a new task called weak-shot learning. It is to learn novel categories by using the knowledge learned in the clean set of base categories. The clean set of base categories refers to the clean, well-labeled image classification dataset.

The proposed method can be summarized into two components. First, the authors train a SimNet on the clean base dataset. It is to learn to capture the semantic concepts from the images. Secondly, the authors conduct supervised classification training on the novel images using similarity labels, where the similarity labels are obtained from the similarity matrix computed by the pre-trained SimNet. To address the label noise in the unseen dataset, sample weighting and graph regularization are employed to better leverage the similarity matrix in training.

**Ethics Review Area:**

["I don’t know"]

**Main Review:**

Strength
- The authors propose an interesting and practical task, and the authors present good baseline to address the problem.
- The idea of similarity transfer is good and reasonable.
- The use of SimNet, and the construction of the relation features is interesting.
- Large number of experiments are conducted to verify different factors in the proposed method.

Weakness
- It is a complicated paper. The paper seems revised many times, but the presentation is not very effective in my point of view. From a high-level view, I think the main goal is to train a novel-category classifier using the regularizations obtained from clean base categories. It might be easier to understand the paper (at least for me, or general audience) by first presenting the overall pipeline and then talk about Sec 4.2. After that, discuss the regularizations terms such as how to obtain the similarity matrix. In addition, in my opinion, the paper is more relevant to web supervision but not zero/few-shot learning. Probably the term "weak-shot learning" is somehow confusing. Moreover, I see the proposed idea fundamentally work for general image classification problem, but the authors tend to restrict the presentation to only fine-grained image classification. To make the paper easier to follow, I would suggest to first address a simple, general problem (general image classification), and then extend it to a more complicated task (fine-grained image classification).
- In current presentation, the authors present a similarity transfer method. One intuitive question is how to apply such method to real-world problems? Is there any restriction? Although the base and novel sets are disjoint in the experiments, one can see the base and novel sets are from the same dataset such as CUB. This implies there are requirements for the base and novel sets. They must belong to the same meta-category, such as bird. If we would like to train a novel bird classifier, but only CompCars dataset is available as the clean base set, I am not sure if the transferability still hold. This is a natural question when we deal with similarity transfer between fine-grained datasets. Instead of dealing with fine-grained dataset, I feel it is more appealing to do general image classification. By doing so, the whole story becomes more elegant, flexible, and easy to sell.
- The proposed task is too complicated. As the results, the authors have to conduct very large amount of experiments to verify many components, and it is very difficult to present all the experiments in the main paper. Instead of simply leaving all the analyses to Appendix, the authors should also summarize the key findings of the experiments in the main paper.

Overall
- The paper addresses an interesting problem, and the proposed method should be technically correct. However, the paper is presenting in a way that is not easy to follow. I think it is a borderline paper. Both the proposed task and method are OK, but it is fine to reject it as well.

****post rebuttal****
Thank you for the response. I would like to keep my rating.

**Time Spent Reviewing:**

10

---

> ### Author Response · Authors · 2021-08-10
> **Answer to Reviewer #2.**
>
> **Q1**. The proposed idea fundamentally work for general image classification problem, but the authors tend to restrict the presentation to only fine-grained image classification. The authors present a similarity transfer method. One intuitive question is how to apply such method to real-world problems? Is there any restriction?
>
> **A1**. As claimed in Line 238-242, we focus on fine-grained classification, because 1) it especially requires expert knowledge to annotate labels for fine-grained data; 2) we conjecture that semantic similarity should be more transferable across fine-grained categories, i.e., the base categories and novel categories belong to the same meta-category. Therefore, we start from transferring across fine-grained categories and extensive experiments have demonstrated the effectiveness of our setting and method. Nevertheless, we have also extended our setting from fine-grained classification to general classification on ImageNet dataset, as claimed in Line 241-242 and reported in Section 5 in Appendix. The reported results indicate that our proposed idea is also effective for the general classification.
>
> ---
> **Q2**. The proposed task is too complicated.
>
> **A2**. The major contribution of this paper is employing transferred similarity to denoise web training data in weak-shot fine-grained classification task, as claimed in Line 72-74. The major finding is the SOTA comparison in Section 5.5 demonstrating the effectiveness of our proposed similarity transfer against baselines. We also introduce several important aspects to support our major finding and explore the proposed learning scenario, including the basic ablation study in Section 5.2, investigating our major idea "similarity transfer" in Section 5.3, analyzing robustness to several factors (Section 5.4, 5.7, 5.8, and 5.9), and a generalized setting in Section 5.10.

---

### Official Review · Reviewer_bmx4 · 2021-07-20

**Rating:** 3
**Confidence:** 5

**Summary:**

The paper is about transferring the semantic information between datasets to learn from the web data. A combination of known techniques such as SimNet [47], the graph regularization [51] and adversarial loss [8] are applied to obtain results.

**Ethical Concerns:**

No concerns

**Ethics Review Area:**

["I don’t know"]

**Limitations And Societal Impact:**

Yes

**Main Review:**

The idea of using web supervision is interesting and there are large-scale web-based datasets available, these are only a few relevant examples: https://paperswithcode.com/dataset/pinterest, https://arxiv.org/pdf/1611.08321.pdf, https://aclanthology.org/2021.naacl-main.473.pdf. The main motivation of the work seems to be able to learn from web data and transfer the leanings to other datasets. I have the following concerns about this work.

- I do not believe the authors have adequately addressed the previous reviews
- From the architectural standpoint, the novelty is very limited. A major contribution could have been a definition of new transfer task from the web data.
- However, in my opinion the authors fail to design the experiments to prove the concept of transferrability from the web data. In particular, I do not believe they use a large-scale web-based dataset as a base and then demonstrate successful transfer to another dataset.
- The paper is not clearly written. The introduction is jumping around zero-/few- shot and web data concepts without ever clearly defining the use case that the authors want to address. In fact, I cannot figure out the use case even after reading the entire article, because the authors claim that they can learn and transfer from the web data, but experiments do not consider a real-life web dataset at all. The Problem Definition section provides a really short and abstract definition of the task. In my opinion a detailed description of the real use case transitioning to its mathematical description would have done a much better job here. The Experiments section has way too many subsections and it is very hard to navigate. There is no clear message in the section and none of the sections really addresses the main question (according to my guess) whether web driven learning is viable in the context of the task that the authors consider. The research questions that the authors want to answer in the section are not defined either, so it is really hard to understand the motivation behind ten different subsections of Section 5. I believe the authors themselves do not see the forest behind the trees and it is even worse for the reader. Please define your research questions and the message you would like to send with your research in a clear and concise way.

Definitely a reject.

**Time Spent Reviewing:**

1 hour

---

> ### Author Response · Authors · 2021-08-10
> **Answer to Reviewer #1.**
>
> **Q1**. The main motivation of the work seems to be able to learn from web data and transfer the leanings to other datasets. In particular, I do not believe they use a large-scale web-based dataset as a base and then demonstrate successful transfer to another dataset.
>
> **A1**. The reviewer may misunderstand our paper. Actually we focus on learning from a set of clean data and transferring to another set of web data as claimed in Line 34-38 or Section 3, instead of learning from web data and transferring to other datasets. Specifically, we explore to learn novel categories from web data with the help of a set of base categories associated with clean data.
>
> ---
> **Q2**. The experiments do not consider a real-life web dataset at all.
>
> **A2**. Actually, we conduct experiments on three real-life fine-grained datasets, including CUB [45], Car [45], and FGVC [19], as described at Section "5.1 Datasets and Implementation". We also conduct experiments on ImageNet as claimed in Line 241-242.
>
> ---
> **Q3**. The experiments section has way too many subsections and it is very hard to navigate.
>
> **A3**. The major experiment is Section 5.5 demonstrating the effectiveness of our proposed similarity transfer against baselines. We also introduce several important aspects to support our major finding and explore the proposed learning scenario, including the basic ablation study in Section 5.2, investigating our major idea "similarity transfer" in Section 5.3, analyzing robustness to several factors (Section 5.4, 5.7, 5.8, and 5.9), and a generalized setting in Section 5.10.

---

> > ### Comment · Reviewer_bmx4 · 2021-08-10
> > **Real-life web dataset**
> >
> > Dear authors, thank you for the clarifications! I believe my main concerns still hold and you have not provided a response specifying ways to address them, namely:
> > - I do not believe that that datasets such as CUB, Car, and FGVC or ImageNet could be classified as web datasets. Therefore I still do not see experiments demonstrating the learning of novel categories from web data.
> > - The overall lack of clarity in the paper is undeniable, experiments section is extremely hard to parse and research questions are unclear, how are you going to address these points (if at all)?

---

> > > ### Author Response · Authors · 2021-08-10
> > > **More clarifications on real-life web dataset and experiments**
> > >
> > > Dear *Reviewer bmx4*, thanks for the supplemental questions.
> > >
> > > 1) As claimed in Line 232-234, the base training/test set and the novel test set are from the original (clean) dataset while **the novel training set is constructed using web images**. More details about constructing web training set for novel categories are described in Line 30-35 in Appendix. Therefore, our overall setting and experiment are "learning from clean data (base categories) and transferring to web data (novel categories)",  as described in Line 34-38 or Section 3.
> > >
> > > 2) Our research question is detailedly described in Section "1 Introduction" and clearly defined in Section "3 Problem Definition". In brief, we explore to learn novel categories from web data with the help of a set of base categories associated with clean data.
> > >
> > > 3) As for the experiment sections, we plan to summarize Section 5.4, 5.7, 5.8, and 5.9 to one section for analyzing the robustness of our method to several factors.

---

> > ### Comment · Reviewer_bmx4 · 2021-08-27
> > **Post rebuttal**
> >
> > I would like to thank the authors for additional clarifications. My concern about web data is cleared to some extent. I advise that the authors move the description of the web data application protocol in the main body, since it's the core of their work. This also raises another concern about reproducibility. How do you intend to release the images scraped from google? Moreover, if you use google search to query images by class label, then your algorithm is essentially supervised by google, no? In my view, the pure web based operation would imply the use of a large collection of images weakly labeled by users via tags/text comments. Additionally, my other concerns such as novelty and clarity of the writing are still not addressed. My opinion is that the paper needs a major revision and its main contribution can only be a web-based processing task in its current form. I think the novelty is still very limited. I will adjust the score accordingly adjusting it up a bit, but essentially the paper id still not good enough to be accepted.

---

> > > ### Author Response · Authors · 2021-08-28
> > > **About Web Data**
> > >
> > > Dear Reviewer *bmx4*, thanks for the additional comments. Scrapping images from Google or Flickr is a conventional practice for webly supervised learning works [52,13,44,12,34]. We could follow previous works and release the scraped images via FTP or Dropbox.

---

### Decision · Program_Chairs · 2021-09-28

**Decision:**

Accept (Poster)

**Comment:**

None of the reviewers recommend accepting this paper.
There was substantive discussion during the rebuttal period, but the reviewers remained of the opinion that the work should not be accepted, with these opinions based on the quality of the presentation, their view of the novelty of the work and the degree to which the experiments demonstrated the utility of the proposed method.
The AC recommends rejection.

**Consistency Experiment:**

NeurIPS has a long history of experimentation. In 2014, NeurIPS ran an experiment in which 10% of submissions were reviewed by two independent committees to quantify the randomness in the review process. This year, we repeated a variant of this experiment to see how the quality of the review process has changed over time.  This paper was part of the experiment and was therefore assigned to two committees (consisting of reviewers, an Area Chair, and a Senior Area Chair) that reached independent decisions.  If both committees made the same recommendation, this recommendation was followed. If a single committee recommended acceptance, the paper was accepted (with the exception of a few cases in which the other committee identified what we considered a fatal flaw, e.g., an error in a key result).

This copy’s committee reached the following decision: **Reject**

The other committee assigned to the paper recommended **Accept (Poster)**.  You can find the other set of reviews, along with any follow up discussion with the authors here:
https://openreview.net/forum?id=vrXuRmaU_jM